# Assessment of Anti-Prostate Cancer Activity among Four Seaweeds, with Focus on *Caulerpa lentillifera* J.Agardh

**DOI:** 10.3390/foods13091411

**Published:** 2024-05-03

**Authors:** Guan-James Wu, Pei-Wen Hsiao

**Affiliations:** 1Department of Food Science, National Penghu University of Science and Technology, Magong 880011, Taiwan; 2Agricultural Biotechnology Research Center, Academia Sinica, Taipei 115201, Taiwan; pwhsiao@gate.sinica.edu.tw

**Keywords:** seaweeds, *Caulerpa lentillifera* J.Agardh, prostate cancer, apoptosis, migration

## Abstract

In response to a global shift towards health-conscious and environmentally sustainable food choices, seaweed has emerged as a focus for researchers due to its large-scale cultivation potential and the development of bioactive substances. This research explores the potential anticancer properties of seaweed extracts, focusing on analyzing the impact of four common edible seaweeds in Taiwan on prostate cancer (PCa) cells’ activity. The study used bioassay-guided fractionation to extract Cl80 from various seaweeds with androgen receptor (AR)-inhibitory activity. Cl80 demonstrated effective suppression of 5α-dihydrotestosterone (DHT)-induced AR activity in 103E cells and attenuated the growth and prostate-specific antigen (PSA) protein expression in LNCaP and 22Rv1 cells. Additionally, Cl80 exhibited differential effects on various PCa cell lines. Concentrations above 5 μg/mL significantly inhibited LNCaP cell proliferation, while 22Rv1 cells were more resistant to Cl80. PC-3 cell proliferation was inhibited at 5 μg/mL but not completely at 50 μg/mL. A clonogenic assay showed that at a concentration of 0.5 μg/mL, the colony formation in LNCaP and PC-3 cells was significantly reduced, with a dose-dependent effect. Cl80 induced apoptosis in all PCa cell types, especially in LNCaP cells, with increased apoptotic cells observed at higher concentrations. Cl80 also decreased the mitochondrial membrane potential (ΔΨm) in a dose-dependent manner in all PCa cell lines. Furthermore, Cl80 suppressed the migration ability of PCa cells, with significant reductions observed in LNCaP, 22Rv1, and PC-3 cells at various concentrations. These compelling findings highlight the promising therapeutic potential of *C. lentillifera* J.Agardh and its isolated compound Cl80 in the treatment of PCa.

## 1. Introduction

With the improvement in people’s living standards worldwide and the rise of the concept of preventive medicine, consumers pay more attention to health care and preservation [1]. Especially in developed or developing countries, the average life expectancy of citizens has increased significantly, resulting in the formation of an aging population structure and a gradual increase in the number of chronic diseases. This seems to lead to another significant worry about the national medical expenditure and social welfare burden [2].

The American Cancer Society estimated that there will be 2.0 million new cancer cases in 2024, including 299,010 new cases of PCa. Among male patients, new cases of PCa, lung cancer, and bowel cancer accounted for 48% of all new cancer cases, with PCa accounting for 29% and ranking first. The probability of death caused by it is about 11%, and the mortality rate ranks second only to lung cancer, which shows the importance of PCa to men’s health [3]. Most PCa is adenocarcinoma, characterized by the massive expression of a prostate-specific antigen (PSA) [4]. And 80% of PCa patients have carcinoma in situ, for which the prognosis is relatively optimistic, and the 5-year life expectancy is about 60–99%. However, 20% of cancer patients have metastatic disease, and the prognosis is not good [5].

Seaweeds are extensively used as food in Asia. Seaweeds can be classed into red (Rhodophyta), green (Chlorphyta), and brown (Phaeophyta) algae based on their pigments. The pigments of red algae are derived from phycoerythrin and phycocyanin; green algae contain chlorophyll a and b; while brown algae contain fucoxanthin and xanthophyll carotenoids [6]. They are botanically classified as marine macroalgae and inhabit the coastal zone. Seaweed contains polysaccharides, lipids, proteins, vitamins, minerals, enzymes, and polyphenols, which are of great nutritional value [7]. Seaweed also has medicinal properties, such as anti-inflammatory, antibacterial, antiviral, anticancer, antidiabetic, hypotensive, hypolipidemic, and neuroprotective activities [8]. Wang et al., demonstrated that sulfated polysaccharides isolated from *Codium fragile* possess anti-inflammatory activity by reducing the production of inflammatory molecules in LPS-induced RAW264.7 cells [9]. In addition to seaweed polysaccharides, numerous studies have identified seaweed-based compounds such as polyphenols, terpenes, fatty acids, and proteins as exhibiting anti-inflammatory properties [10]. Polyphenols, known for their antioxidant activity, also possess anticancer [11] and antiviral capabilities [12]. In terms of anticancer activity, seaweed polysaccharides have to be mentioned. Sulfated galactans from red algae, ulvan from green algae, and fucoidan from brown algae have been reported to exhibit anticancer activity. In red algae, porphyran and agar extracted from *Porphyra yezoensis* and *Kappaphycus striatum*, respectively, demonstrated inhibitory effects on the growth of cancer cells including Hep3B, HepG2, MCF-7, K562, and HT-29 [13]. In green algae, Pradhan et al., observed that ulvan from *Ulva lactuca* exhibited significant cytotoxic activity against HepG2, MCF-7, and HeLa cancer cells [14]. Fucoidan, extensively studied for its anticancer properties in brown algae, has shown promising inhibitory effects on the growth of various human cancer cells, with its efficacy confirmed in animal models [15]. This shows that seaweeds can be a new source of discovery of biological activity.

Recently, people’s food consumption patterns have shifted towards health-consciousness, while at the same time caring about environmental and sustainability issues [16]. Seaweed has attracted the attention of researchers in the past few decades because it can be cultured on a large scale, which contributes to environmentally sustainable operations [17] and provides many opportunities for the development of bioactive substances [8]. Seaweed has been extensively studied for its anticancer activity. However, most of the research on its active ingredients focuses on polysaccharides, and there are relatively few anticancer studies on small molecule extracts [10]. Therefore, this study focused on the analysis of anticancer research on four common edible seaweeds with sustainable management benefits in Taiwan and observed their effects on the activity of PCa cells. We used 103E cell-assisted bioactivity-guided fractionation to isolate seaweed extracts with AR-inhibitory activity. The extract was evaluated for its different effects on various PCa cell lines, such as LNCaP, 22Rv1, and PC-3 cells, including PSA secretion, cell growth and apoptosis, and cell migration.

## 2. Materials and Methods

### 2.1. Materials and Chemicals

The seaweed powders of *Porphyra dentata* Kjellman, *Monostroma nitidum* Wittrock, *Sargassum cristaefolium* C.Agardh, and *Caulerpa lentillifera* J.Agardh were obtained from Everyone Excellent-Algae (Penghu, Taiwan). The RPMI-1640 (Roswell Park Memorial Institute-1640, Buffalo, NY, USA) medium, DMED (Dulbecco’s modified Eagle’s medium), fetal bovine serum (FBS), and charcoal-coated dextran-stripped FBS were purchased from Gibco (Waltham, MA, USA). The acetic acid, anhydrous absolute ethanol, and 96% ethanol of analytical grade were purchased from PanReac (Barcelona, Spain). Cosmosil 75C_18_ was purchased from Nacalai Tesque (Kyoto, Japan). 5α-dihydrotestosterone (DHT), 3-(4,5-Dimethylthiazol-2-yl)-2,5-diphenyltetrazolium bromide (MTT), dimethyl sulfoxide (DMSO), crystal violet, glutaraldehyde, Rhodamine-123 (Rh-123), and other general chemicals were obtained from Sigma-Aldrich (St. Louis, MO, USA).

### 2.2. Seaweed Extract Preparation

Four kinds of dried seaweed powders of *P. dentata* Kjellman, *M. nitidum* Wittrock, *S. cristaefolium* C.Agardh, and *C. lentillifera* J.Agardh were macerated separately in 95% ethanol at room temperature. After 48 h, the crude extract was taken out, and a new ethanol solvent was added. This extraction was repeated three times in total. The collected crude extracts were mixed and filtered through filter paper. Ethanol in the crude extract was removed by using a low-pressure rotary evaporator at 40 °C. Finally, the crude extract was dried using a freeze-dryer. The lyophilized crude extracts of *P. dentata* Kjellman, *M. nitidum* Wittrock, *S. cristaefolium* C.Agardh and, *C. lentillifera* J.Agardh seaweeds are represented by the codes Pd, Mn, Sc, and Cl, respectively, and stored in a −20 °C refrigerator until further use.

Crude seaweed extracts (samples such as Pd, Mn, Sc, and Cl) were subjected to column fractionation on 75 μm C18-Reversed phase silica gel (Cosmosil 75C_18_). They were eluted with 5 bed volumes of 20%, 40%, 60%, 80%, or 100% ethanol (*v*/*v*)—the fractions were denoted as sample20, sample40, sample60, sample80, and sample100, respectively. All these extracts were concentrated at 40 °C under reduced pressure and dried using a freeze-dryer. The fractions were stored at –80 °C until further use.

### 2.3. Cell Lines and Culture

The cell cultures of human prostate cancer LNCaP, 22Rv1, and PC-3 cells were obtained from ATCC (American Type Culture Collection, Manassas, VA, USA). 22Rv1-derived 103E cells were provided by Dr. Hsiao (Agricultural Biotechnology Research Center, Academia Sinica, Taipei, Taiwan). LNCaP and PC-3 cells were maintained in DMEM, while 22Rv1- and 22Rv1-derived 103E cells were maintained in RPMI-1640. Both media were supplemented with 10% FBS and incubated at 37 °C in a humidified atmosphere of 5% carbon dioxide (CO_2_). The cells were routinely inspected microscopically for stable phenotypes.

### 2.4. Luciferase Assay

Luciferase assay was performed as described by Lin et al., (2007) [18] to determine the effect of seaweed fractions on the androgen receptor (AR) function in the 22Rv1-derived 103E cells. Briefly, 2 × 10^4^ cells were seeded in a 96-well plate with an RPMI 1640 medium containing 5% charcoal-coated dextran-stripped fetal bovine serum for 24 h and then treated with different concentrations of seaweed extracts or fractions. After the treatment for 20 h, the cells were lysed with a commercial passive lysis buffer (Promega, Madison, WI, USA). The luciferase activity in a cell lysate was measured using a luciferase assay system (Promega, Madison, WI, USA) according to the manufacturer’s protocol and normalized to the protein concentration measured by a Coomassie protein assay kit (Thermo Fisher Scientific, Waltham, MA, USA). The inhibition of AR by each treatment was calculated by the relative luciferase activity induced by 10 nM DHT as 0% inhibition and vehicle as 100% inhibition.

### 2.5. Cell Proliferation Assay (MTT Assay)

Cell proliferation was assessed by MTT assay [19], and this assay is based on the observation that live cells possess mitochondria with active enzymes that are capable of reducing MTT to a dark blue visible reaction product form. Briefly, the cells were seeded in a 96-well plate and incubated at 37 °C in a humidified atmosphere of 5% carbon dioxide (CO_2_) for 24 h. Different doses of seaweed extracts or fractions were tested on different cell lines, while the 0.1% DMSO-treated cells were used as the vehicle control. After the treatment for 48 h, the MTT was added and incubated for 4 h at 37 °C. The resulting formazan crystals were dissolved in DMSO, and the absorbance was recorded at 570 nm using a microplate spectrophotometer (Epoch, BioTek Instrument, Inc., Winooski, VT, USA). The results of the cell proliferation assay were expressed as the cell viability percentage against the vehicle control group, which was calculated using the formula [(Abs_(570 nm)_ of the treated sample/Abs_(570 nm)_ of vehicle control)] × 100%.

### 2.6. Prostate-Specific Antigen (PSA) 

The cells were seeded in a 96-well plate with RPMI 1640 medium containing 5% charcoal-coated dextran-stripped fetal bovine serum for 24 h and then treated with different doses of seaweed extracts or fractions and 1 nM DHT. After the treatment for 24 h, the PSA levels in the cell culture supernatant were quantified using a commercial human KLK3/PSA Immunoassay kit (R&D Systems, Minneapolis, MN, USA) as directed by the manufacturer [20].

### 2.7. Colony-Forming Assay

The cells were seeded at a density of 5 × 10^3^ cells per well in a 24-well plate for 24 h. Post-incubation, the cells were treated with different concentrations of Cl80 fraction. Post-treatment, the cells were allowed to grow in culture media with repeated changes of fresh media after 3 days for up to 12 days. On the 12th day, the cells were stained with 0.1% (*w*/*v*) crystal violet after being fixed in ice-cold 1% glutaraldehyde. The cells stained with crystals were then extracted with 20% acetic acid and quantitatively measured by absorbance at 595 nm [18]. The relative colony-forming growth of all treatments was analyzed and presented as a percentage of inhibition compared with the control treatment of the same cell line.

### 2.8. Apoptosis Determination 

The cells were seeded at a density of 2 × 10^5^ cells per well in a 6-well plate for 24 h. Post-incubation, the cells were treated with different concentrations of Cl80 fraction for 24 h. After treatment, the cells were harvested, washed twice with PBS, and subsequently re-suspended in a binding buffer containing FITC-conjugated Annexin-V and PI in the dark for 15 min. After incubation, apoptotic cells were analyzed by flow cytometry (FACSCantoTM, BD Biosciences, Milpitas, CA, USA). A minimum of 10,000 cells per sample were counted and analyzed using FlowJo^TM^ v7.6.5 software [21].

### 2.9. Change in Mitochondrial Membrane Potential (ΔΨm)

The cells were seeded at a density of 2 × 10^5^ cells per well in a 6-well plate for 24 h. Post-incubation, the cells were treated with different concentrations of Cl80 fraction for 24 h. After treatment, the cells were collected and centrifuged and re-suspended in 500 μL of PBS. Following incubation with Rh-123 (10 μM) for 10 min in the dark at room temperature, the fluorescence intensity was measured by flow cytometry (FACSCantoTM, BD Biosciences, Milpitas, CA, USA). A minimum of 10,000 cells per sample were counted and analyzed using FlowJo^TM^ v7.6.5 software [22].

### 2.10. Wound Healing Assay

The wound was linear and produced using a two-well Ibidi silicone culture insert (Ibidi GmbH, Martinsried, Germany) following the instructor’s recommendations. Briefly, the cell culture inserts were transferred to 24-well plates. Then, 70 μL containing 3 × 10^4^ cells was added into each insert well and incubated at 37 °C and 5% CO_2_. After 24 h, inserts were removed, creating a 500 μm wide gap between cells in each plate well. Then, the cells were treated with different concentrations of Cl80 fraction. After incubation, the cells were imaged, and the area of the cell-free region was measured using ImageJ v1.53t software [23]. The migration ratio was determined by the following equation:Migration ratio = [1 − (D_48(/24)_/D_0_)] × 100%
where D_0_ and D_48(/24)_ represent the wound area on day 0 and after 48 h (or 24 h), respectively.

### 2.11. Statistical Analysis

The results are presented as the mean ± SD. Data were analyzed using the statistical analysis system (SAS) program. An unpaired *t*-test was used to determine the significance of differences between two groups, and a one-way analysis of variance (ANOVA) was used to estimate the differences between three or more groups. In all cases, a *p* < 0.05 was considered statistically significant.

## 3. Results

### 3.1. Bioactivity-Guided Fractionation of Seaweeds

In this study, 103E cells with androgen-induced PSA-Luc activity were used as the target cells for AR activity analysis, and the cells were provided by Dr. Hsiao. Dr. Hsiao’s team transformed a PSA-Luciferase reporter gene into 22Rv1 cells and obtained a stable 103E cell line [18]. Using this cell line with chromatography technology as a screening platform, the effects of four kinds of seaweed on the AR suppression in 103E cells were analyzed. The result is shown in Figure 1, and all the crude seaweed extracts had no significant inhibitory effect on AR activity in 103E cells. However, Pd60, Pd80, and Pd100 in the chromatographic fraction of *P. dentata* Kjellman crude seaweed extract (Pd) could significantly inhibit the AR activity in DHT-stimulated 103E cells. Among them, Pd80 had the best inhibitory effect. At a concentration of 50 μg/mL, the AR suppression of Pd80 was 71.03 ± 3.51%, and its IC50 was about 32.96 μg/mL. In the chromatographic fractions of *M. nitidum* Wittrock crude seaweed extract (Mn), except for Mn80 and Mn100, which at a high concentration (50 μg/mL) could inhibit the AR activity in DHT-stimulated 103E cells by 40.22 ± 7.53% and 56.20 ± 13.40%, the rest had no inhibitory effect. In the chromatographic fractions of *S. cristaefolium* C.Agardh crude seaweed extract (Sc), Sc80 and Sc100 could significantly inhibit the AR activity in DHT-stimulated 103E cells. Among them, Sc80 had the best inhibitory effect. At a concentration of 50 μg/mL, Sc80 could effectively inhibit the AR activity in DHT-stimulated 103E cells by 59.50 ± 12.00%, and its IC50 was about 34.21 μg/mL. The chromatographic fractions of *C. lentillifera* J.Agardh crude seaweed extract (Cl) showed that Cl60, Cl80, and Cl100 could significantly inhibit the AR activity in DHT-stimulated 103E cells. Among them, Cl80 had the best inhibitory effect. At a concentration of 50 μg/mL, Cl80 could effectively inhibit the AR activity in DHT-stimulated 103E cells by 99.17 ± 18.32%, and its IC50 was only 0.92 μg/mL. According to the above research results, it is shown that some seaweed natural products have the ability to inhibit the AR activity in 103E cells that are stimulated by an androgen hormone. In the case of Cl80, whose IC50 was only 0.92 μg/mL, when further observed under the microscope, it was found that Cl80 could obviously cause morphological changes in 103E cells, showing that it has obvious physiological effects on 103E cells.

### 3.2. Effect of Seaweed Extracts on LNCaP Cell Proliferation

To observe the effect of a seaweed extract on the proliferation of androgen-dependent PCa cells, in this study, LNCaP cells were used as target cells to analyze the effect of crude seaweed extracts and their chromatographic fractions on LNCaP cells. The results are shown in Figure 2: the seaweed-80 fractions obtained by chromatography had an obvious inhibitory effect on the growth of LNCaP cells. At a concentration of 50 μg/mL, the inhibitory effects of Pd80, Mn80, Sg80, and Cl80 on the growth of LNCaP cells were 35.48 ± 6.96%, 61.85 ± 3.26%, 48.94 ± 8.96%, and 7.76 ± 2.06% of the control group, respectively. Especially for the fraction Cl80, its IC50 was only 5.79 μg/mL. Moreover, Pd80 and Sg80 also had good inhibitory abilities, and their IC50 values were 22.4 μg/mL and 43.3 μg/mL, respectively, while the rest of the seaweed fractions had no obvious inhibitory ability.

### 3.3. Effect of Cl80 on PSA Expression

The above results show that Cl80 has better AR inhibitory activity and also significantly inhibited the androgen-dependent LNCaP cell growth. Therefore, this study focused on Cl80 and further analyzed its effect on androgen-induced PSA production in LNCaP and 22Rv1 cells. The effect of Cl80 on the androgen-induced PSA production in LNCaP and 22Rv1 cells was further analyzed. The results are shown in Figure 3: Cl80 can effectively inhibit DHT to stimulate LNCaP and 22Rv1 cells to secrete PSA in a dose-dependent manner. Especially for LNCaP cancer cells, its IC50 is <1 μg/mL, while the IC50 of Cl80 for 22Rv1 cancer cells is between 1 and 10 μg/mL. It has been shown that Cl80 can effectively inhibit the activity of PCa cells activated by androgen.

### 3.4. Effect of Cl80 on PCa Cell Proliferation and Development

To analyze the effect of Cl80 on the survival and development of different types of PCa cells, the selected PCa cell lines included LNCaP, 22Rv1, and PC-3, which can be divided into androgen-dependent and non-androgen-dependent types. In terms of the short-term effects of Cl80 on PCa proliferation (Figure 4), 0.5 μg/mL of Cl80 could not effectively inhibit the growth of all types of PCa cells. Cl80, at a concentration above 5 μg/mL, significantly inhibited the growth of LNCAP in a dose-dependent manner. At 50.0 μg/mL, the effect of Cl80 on the proliferation of LNCaP cells was 24.39 ± 1.59% that of the control group. 22Rv1 cancer cells were more resistant to Cl80. When the concentration increased to 25.0 μg/mL and 50.0 μg/mL, the effects of Cl80 on the proliferation of 22Rv1 cells were 70.65 ± 2.41% and 59.97 ± 1.86% that of the control group, respectively. As for its effect on PC-3 cancer cells, Cl80 at a concentration of 5.0 μg/mL could significantly inhibit the proliferation of PC-3 cancer cells, reaching 65.07 ± 1.70% of that of the control group. However, when the concentration of Cl80 increased to 50.0 μg/mL, its inhibitory effect on PC3 growth remained 44.86 ± 0.47%. It is shown that Cl80 does not seem to be able to completely inhibit the proliferation of PC-3 cells.

Subsequently, clonogenic assays were performed to assess the long-term effects of Cl80 on the development of human PCa cells LNCaP, 22Rv1, and PC-3. The results are shown in Figure 5: 0.50 μg/mL of Cl80 could significantly decrease the colony formation in LNCaP and PC-3 cells. After 12 days of treatment with 1.00 μg/mL Cl80, the colony formation in LNCaP was reduced to 10.76 ± 1.51% compared to that of the control group. With a continued increase in the concentration of Cl80 beyond 5.00 μg/mL, the colony formation of LNCaP cancer cells was less than 2.0% of that of the control group (Figure 5A). This shows that Cl80 has an excellent inhibitory effect on the development of androgen-dependent LNCaP prostate cancer cells. In addition, 1.00 μg/mL Cl80 reduced the colony formation of PC-3 cells to 7.87 ± 0.90% of the control. However, even if the concentration of Cl80 was increased to 25.00 μg/mL, the colony formation of PC-3 cancer cells was still 3.48 ± 0.75% of that of the control (Figure 5C). Also, 25.00 μg/mL of Cl80 decreased the colony formation of 22Rv1 cancer cells, which was 0.91 ± 0.41% of that of the control (Figure 5B).

### 3.5. Effect of Cl80 on LNCaP Cell Apoptosis

An Annexin V/PI assay was conducted to determine the cell death type. Cl80 induced apoptosis in all types of prostate cancer cells (Figure 6), especially in androgen-dependent LNCaP prostate cancer cells. Apoptotic cells were significantly increased after treatment with 10 μg/mL Cl80 (19.70 ± 1.25%), and the effect increased with the increase in the concentration. When LNCaP cells were treated with 50 μg/mL Cl80, the percentage of apoptotic cells was increased to 27.50 ± 1.21%. Similar results were also found in 22Rv1 with AR receptor mutations and PC-3 prostate cancer cells lacking AR receptors; 50 μg/mL of Cl80 could cause 14.24 ± 1.06% and 16.04 ± 1.12% of cell apoptosis, respectively, and also had a dose-dependent manner.

To investigate the likely involvement of the mitochondrial pathway in CL80-induced apoptosis, PCa cells were stained with the fluorescent mitochondrial probe Rh-123. We demonstrated that ΔΨm decreased in PCa cells treated with Cl80 (Figure 7). Cl80 at 1 μg/mL reduced the fluorescence intensity of Rh-123 in LNCaP cells by 4.73 ± 0.68%, which was higher than that of the control group (2.83 ± 0.67%). When the concentration of Cl80 was 50 μg/mL, it reduced the fluorescence intensity of Rh-123 in LNCaP cells by 24.63 ± 2.46%. Similar results can also be found in 22Rv1 and PC-3 cancer cell lines. At a concentration of 50 μg/mL, Cl80 can reduce the fluorescence intensity of Rh-123 in 22Rv1 and PC-3 cells by 12.43 ± 0.81% and 23.33 ± 2.18%, respectively.

### 3.6. Effect of Cl80 on PCa cell Migration

An in vitro wound healing assay was conducted to assess the impact of the Cl80 on the cell migration ability of human PCa cells LNCaP, 22Rv1, and PC-3. A gap of 500 μm width was created in the cell culture dish using a commercially available culture insert. PCa cells were treated with different concentrations of Cl80, and the effects of Cl80 on the migration ability of PCa cells were observed (Figure 8). After 48 h of cell culture, the LNCaP cells in the control group could seal the 500 μm gap but not completely fill it, and the cell migration ratio was 71.85 ± 6.16%. After 0.5 μg/mL Cl80 treatment of LNCaP cells, the cell migration ratio decreased to 53.10 ± 2.55%. As the concentration of Cl increased, the migration ratio of the LNCaP cells was significantly suppressed. Similar results were also found in 22Rv1 and PC3 cells. Additionally, 20 μg/mL Cl80 significantly reduced the migration ratio of 22Rv1 and PC-3 cells by 27.76 ± 0.88% and 10.30 ± 2.03%, respectively.

## 4. Discussion

Bioassay-guided fractionation of seaweed extracts associated with chromatographic separation techniques can improve the efficiency of isolating bioactive substances [24]. Therefore, this study used the PSA gene expressed by 103E cells as a biological activity indicator, combined with MPLC technology, to extract extracts with anti-PCa cell activity potential from four types of seaweed. The crude extracts of the four types of seaweed demonstrated a mild inhibitory effect on AR activity. In this study, Cl80 with AR inhibitory activity was quickly extracted from various seaweeds through a bioassay-guided fractionation. (Figure 1). Cl80 not only effectively suppressed DHT-induced AR activity in 103E cells but also attenuated the growth and expression of PSA in LNCaP cells (Figure 2 and Figure 3).

In general, AR serves as an androgenic hormone receptor, playing a pivotal role in regulating the differentiation and proliferation of prostate epithelial cells [25]. The initiation and progression of PCa are intricately linked to the AR signaling pathway [26]. Consequently, the early treatment of PCa capitalized on the disease’s dependence on AR activity [27]. Therefore, androgen deprivation therapy (ADT), which includes surgery or chemical methods to block the action of the AR, was employed to control the progression of the disease [28]. The expression of PSA is reliant on androgen signaling within prostate epithelial cells and has been widely utilized as a marker for prostate cancer growth [29]. Moreover, LNCaP cells exhibit a dependence on and responsiveness to androgens, requiring androgens in the medium for their growth [30,31].Therefore, the effect of Cl80 with AR inhibitory activity of inhibiting the growth of LNCaP is expected (Figure 4).

While ADT effectively manages the progression of PCa, the response to ADT diminishes over time, leading to the emergence of castration-resistant prostate cancer. Cancer cells can restore AR signaling through various mechanisms, including intracrine androgen synthesis, AR overexpression and amplification, point mutations, acquisition of constitutively active AR splice variants, and dysregulation of AR coactivators/corepressors that sensitize AR in response to ligand binding [27]. These alterations are not mutually exclusive and may coexist in the same patient [32]. This increases the difficulty of treatment and is ultimately the cause of death of PCa patients. 22Rv1 cells are androgen-independent prostate cancer cells. Due to mutations in their AR, these cells can be continuously activated and maintain the physiological functions of prostate cancer cells even in the absence of androgen [33]. Cl80 can inhibit the growth and PSA production in 22Rv1 cells, although the effect is slightly inferior to its performance in LNCaP (Figure 3 and Figure 4). Additionally, Cl80 can significantly inhibit the growth of PC3 cells (Figure 4). Although PC-3 cancer cells are also androgen-independent prostate cancer cells, they lack AR, and their growth is completely unaffected by androgen [34]. Recent studies have pointed out that PC3 cells possess similar characteristics to cancer stem cells [35].

A colony-forming assay was employed to assess the impact of Cl80 on the growth and development of different types of PCa cells. A colony-forming assay is an in vitro cell survival assay based on the ability of single cells to grow into colonies. For cancer cells, this assay serves to delineate the progression of a singular cancer cell into a tumor, simulate the capacity of cancer cells to disseminate and give rise to another distinct tumor entity, and illustrate the proliferative capabilities of cancer cells [36,37]. In this study, Cl80 exhibited a pronounced inhibitory effect on the colony formation of PCa cells, particularly evident in PC3 cells, with an IC50 for Cl80 below 0.5 μg/mL (Figure 5). Despite increasing the concentration to 25 μg/mL, Cl80 proved incapable of fully suppressing the growth of PC3 cells. After 12 days of treatment at this concentration, only 2.8 ± 0.2% of PC3 cells remained viable. Whether this phenomenon can be attributed to the characteristics of cancer stem cells, making them more resistant to the cytotoxicity of Cl80, requires further investigation. It is noteworthy that numerous studies have highlighted the inherent resistance of cancer stem cells to anticancer drugs [38].

Cl80 inhibited the proliferation of PCa cell lines in a dose-dependent manner, with more selectivity toward LNCaP cells. Treatment with Cl80 induced apoptosis in PCa cell lines, as evidenced by a dose-dependent increase in the annexin V positivity (Figure 6) and disruption of mitochondrial membrane potential (Figure 7). Cancer cells are susceptible to dysregulated apoptosis, and any drug possessing the potential to induce apoptosis can be regarded as a novel candidate for cancer treatment [39]. During apoptosis, a key event involves the translocation of membrane phosphatidylserine (PS) from the inner side of the plasma membrane to the outer surface of the cell. Annexin V, binding with high affinity to exposed PS on the apoptotic cell surface, serves as an indicator of apoptosis [40,41]. In the regulation of apoptosis in mammalian cells, mitochondria play a crucial role. The early stages of apoptosis are consistently accompanied by the disruption of the mitochondrial membrane potential. Subsequent to the loss of mitochondrial membrane potential, specific proteins in the mitochondrial intermembrane space, such as cytochrome c—the second mitochondria-derived activator of caspases—and apoptosis-inducing factor, are released into the cytoplasm, thereby activating the apoptotic process. Eventually, DNA fragmentation occurs, which is a typical phenomenon of cell apoptosis [42].

Metastasis, a phenomenon in which cancer cells separate from the primary tumor, traverse interstitial tissues, and eventually establish themselves in distant organs, is accountable for roughly 90% of cancer-related fatalities [43]. Cellular migration plays a pivotal part in the process of metastasis and infiltration [44]. And our study has proven that Cl80 can significantly suppress cellular migration in LNCaP, 22Rv1, and PC-3 cells. The inhibitory effect of Cl80 on the migration of LNCaP cells was better than that of 22Rv1 cells. Both LNCaP and 22Rv1 cells tend to form cohesive monolayers, which move collectively as clusters or by sliding along their neighbors [45]. Kim’s research demonstrates that bicalutamide, an AR antagonist, effectively reversed DHT-induced mobility in LNCaP cells [46]. This finding substantiates the involvement of the AR signaling pathway in the migration of LNCaP cells. Similarly, Cl80 appears to impede the migration of LNCaP cells by inhibiting the AR signaling pathway. Consequently, Cl80 exhibits a limited efficacy in inhibiting the migration of 22Rv1 cells. As PC3 cells migrate, the often form temporary voids through which other cells can migrate [45]. Intriguingly, Cl80 demonstrates a significant inhibitory effect on the migration of PC3 cells, surpassing its impact on 22Rv1 cells. The morphology of PC3 cells has a variety of spread areas and different geometries. These results suggest that Cl80 may harbor a not-yet-unidentified mechanism of action contributing to its efficacy in inhibiting PCa cell migration.

Cl80 showed the ability to inhibit the activity of PCa cells in this study, and its effect will also vary depending on the PCa cell line. However, there is still a long way to go before Cl80 can become a real drug. The biological variability in Cl80 between individual cells and patient samples may affect treatment efficacy. To address this issue, future studies could explore variability in response to Cl80 treatment by using cell lines derived from different patients, or even performing in vivo studies to evaluate the effects in a more physiologically relevant context. Furthermore, studying potential biomarkers or molecular signatures associated with Cl80 responses could provide insight into underlying mechanisms and help identify subpopulations of patients who may benefit the most from this treatment.

## 5. Conclusions

The present study highlights the superior anti-PCa activity of *C. lentillifera* J.Agardh among four types of seaweed. The isolation of Cl80 from *C. lentillifera* J.Agardh using bioassay-guided fractionation demonstrated a significant inhibition of the PSA expression in PCa cells. Furthermore, Cl80 induced apoptosis, effectively inhibiting the growth and colony formation of various PCa cell lines, with notable efficacy against androgen-sensitive LNCaP cells. Additionally, Cl80 exhibited a substantial inhibitory effect on cell migration, suggesting its potential utility in adjuvant therapy of prostate cancer. These findings underscore the promising prospects of *C. lentillifera* J.Agardh and its isolated compound Cl80 as potential candidates for the treatment of PCa.

## Figures and Tables

**Figure 1 foods-13-01411-f001:**
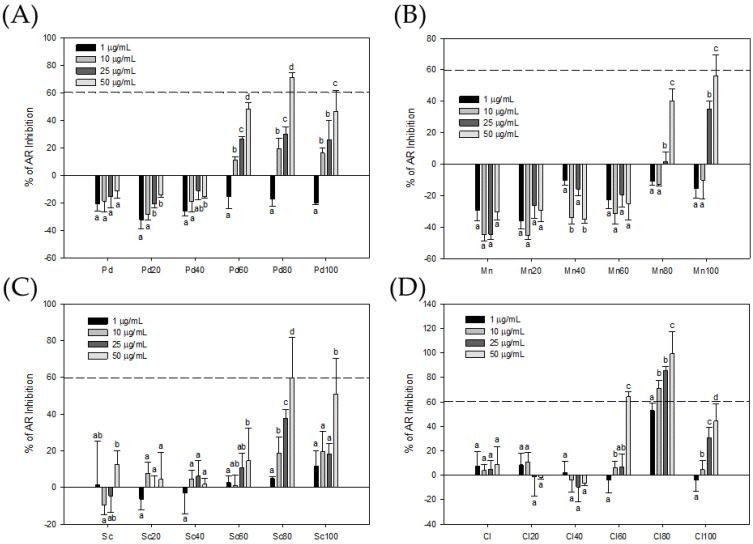
Bioactivity-guided fractions of AR inhibitors in crude extract (Pd) and MPLC fractions (Pd20, Pd40, Pd60, Pd80, and Pd100) of *P. dentata* Kjellman (**A**); crude extract (Mn) and MPLC fractions (Mn20, Mn40, Mn60, Mn80, and Mn100) of *M. nitidum* Wittrock (**B**); crude extract (Sc) and MPLC fractions (Sc20, Sc40, Sc60, Sc80, and Sc100) of *S. cristaefolium* C.Agardh (**C**); and crude extract (Cl) and MPLC fractions (Cl20, Cl40, Cl60, Cl80, and Cl100) of *C. lentillifera* J.Agardh (**D**). Each value presents the mean ± SD of three independent experiments. Different letters indicate significant differences (*p* < 0.05) between different concentrations in different samples.

**Figure 2 foods-13-01411-f002:**
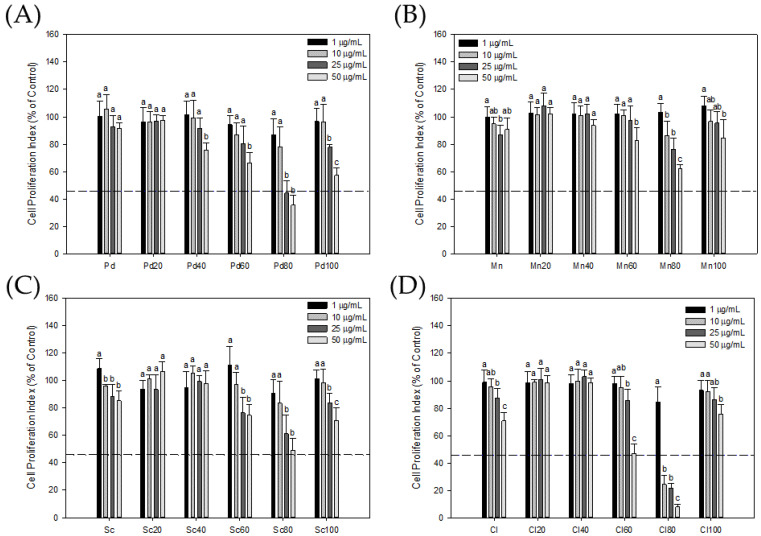
The seaweed extracts inhibited the viability of LNCaP cells. The cytotoxicity of the crude extract (Pd) and MPLC fractions (Pd20, Pd40, Pd60, Pd80, and Pd100) of *P. dentata* Kjellman (**A**); the crude extract (Mn) and MPLC fractions (Mn20, Mn40, Mn60, Mn80, and Mn100) of *M. nitidum* Wittrock (**B**); the crude extract (Sc) and MPLC fractions (Sc20, Sc40, Sc60, Sc80, and Sc100) of *S. cristaefolium* C.Agardh (**C**); and the crude extract (Cl) and MPLC fractions (Cl20, Cl40, Cl60, Cl80, and Cl100) of *C. lentillifera* J.Agardh (**D**) for LNCaP cells were assessed by MTT assay. Each value presents the mean ± SD of three independent experiments. Different letters indicate significant differences (*p* < 0.05) between different concentrations in different samples.

**Figure 3 foods-13-01411-f003:**
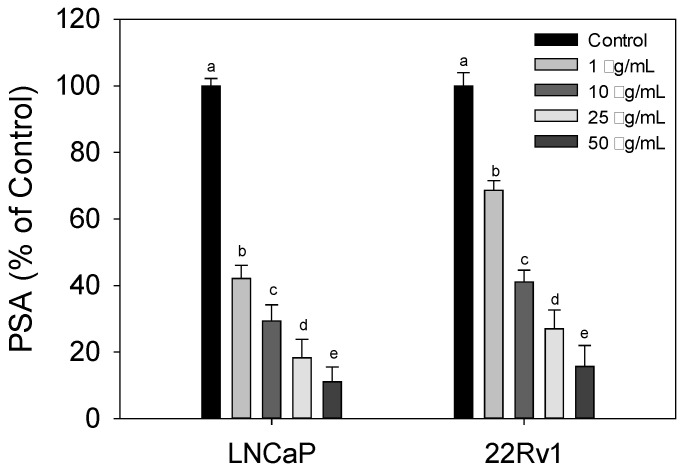
Cl80 suppressed the secretion of PSA in the presence of DHT in LNCaP and 22Rv1 cells. The data are presented as the mean ± SD of three independent experiments. Different letters indicate significant differences (*p* < 0.05) between different sample concentrations in the same cell line.

**Figure 4 foods-13-01411-f004:**
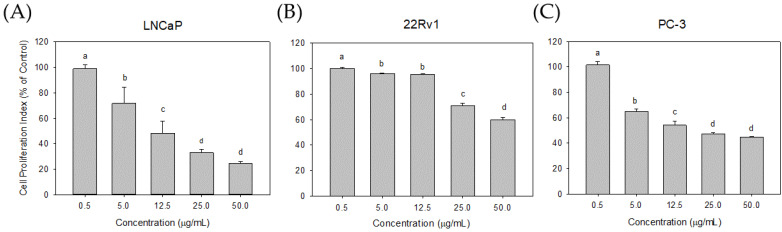
Cl80 inhibited the viability of LNCaP (**A**), 22Rv1 (**B**), and PC-3 (**C**) cells. The cell viability of PCa cells treated with Cl80 exhibited a significant decrease in a dose-dependent manner. The data are presented as the mean ± SD of three independent experiments. Different letters indicate significant differences (*p* < 0.05).

**Figure 5 foods-13-01411-f005:**
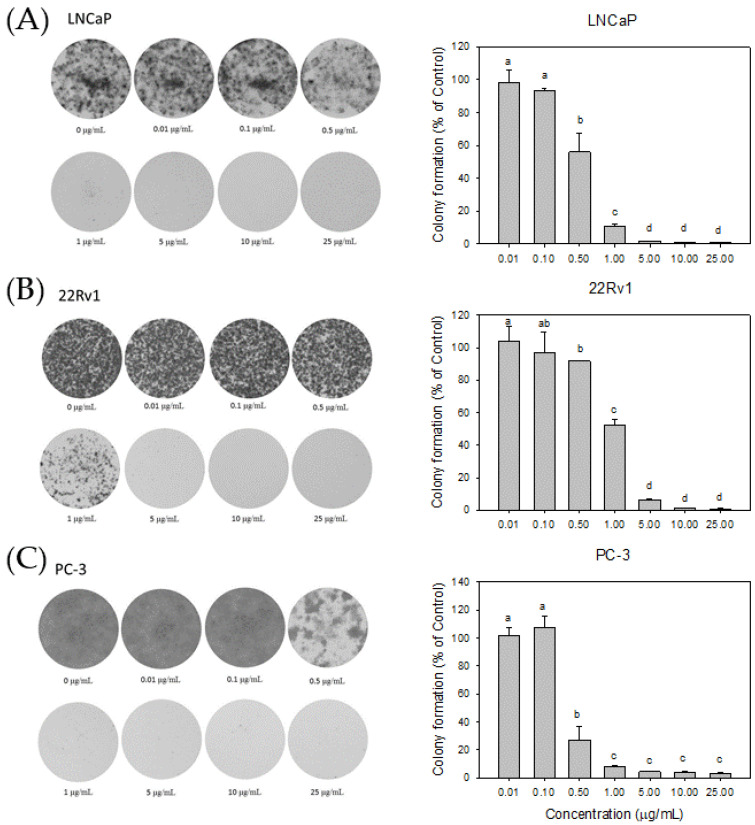
Cl80 decreased the colony formation in LNCaP (**A**), 22Rv1 (**B**) and PC-3 (**C**) prostate cancer cells. The data are presented as the mean ± SD of three independent experiments. Different letters indicate significant differences (*p* < 0.05).

**Figure 6 foods-13-01411-f006:**
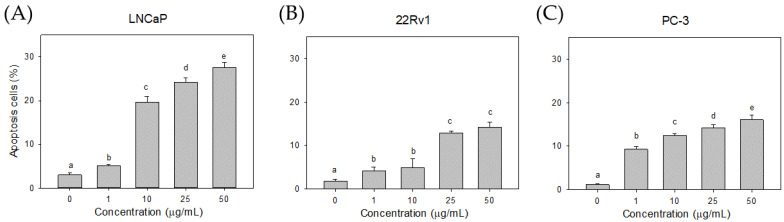
Effects of Cl80 on apoptosis of LNCaP (**A**), 22Rv1 (**B**), and PC-3 (**C**) prostate cancer cells. The data are presented as the mean ± SD of three independent experiments. Different letters indicate significant differences (*p* < 0.05).

**Figure 7 foods-13-01411-f007:**
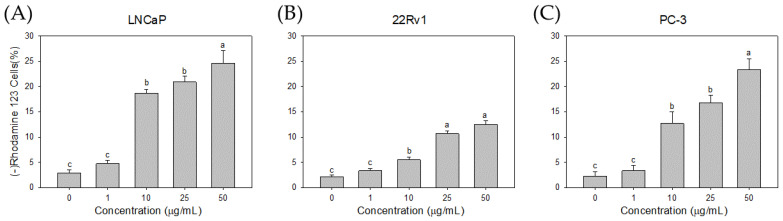
Effects of Cl80 on mitochondrial membrane potential of LNCaP (**A**), 22Rv1 (**B**), and PC-3 (**C**) prostate cancer cells. The data are presented as the mean ± SD of three independent experiments. Different letters indicate significant differences (*p* < 0.05).

**Figure 8 foods-13-01411-f008:**
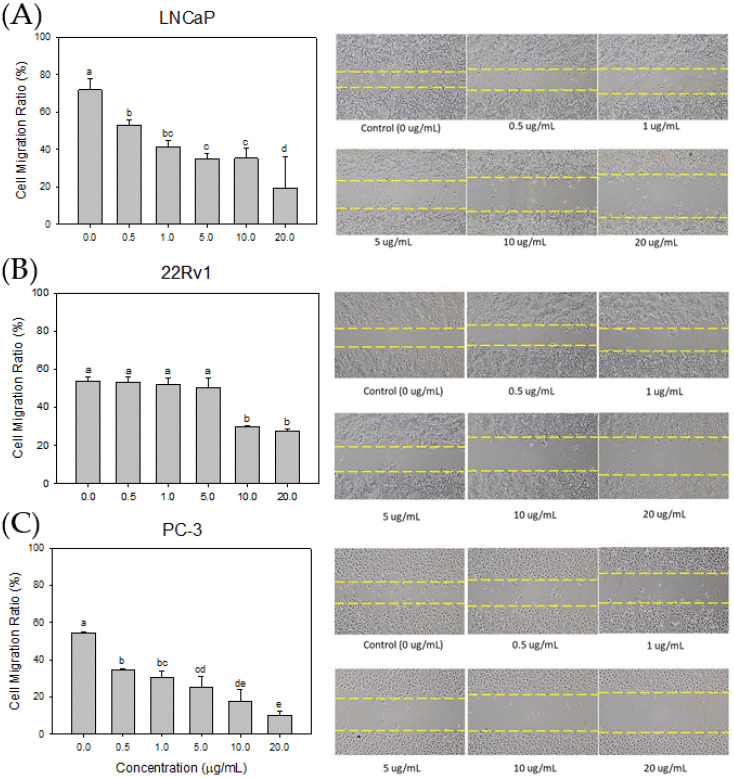
Effects of Cl80 on the metastasis (spreading) ability of LNCaP (**A**), 22Rv1 (**B**), and PC-3 (**C**) prostate cancer cells. The data are presented as the mean ± SD of three independent experiments. Different letters indicate significant differences (*p* < 0.05).

## Data Availability

The original contributions presented in the study are included in the article, further inquiries can be directed to the corresponding author.

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
