# Peer review of "Assessment of Anti-Prostate Cancer Activity among Four Seaweeds, with Focus on Caulerpa lentillifera J.Agardh"

_foods, 2024, doi:10.3390/foods13091411_

Round 1

Reviewer 1 Report

Comments and Suggestions for Authors

The study on the anticancer properties of seaweed extracts, particularly Cl80, for prostate cancer has several limitations:

The study primarily relies on in vitro cell line models (LNCaP, 22Rv1, PC-3) to assess the anticancer properties of Cl80. While these models provide valuable insights, they may not fully replicate the complexity of human prostate cancer in vivo. It is not clear on what basis authors have selected Cl80 alone for testing.

While the study demonstrates the efficacy of Cl80 in inhibiting AR activity, inducing apoptosis, and inhibiting cell proliferation in prostate cancer cell lines, the exact molecular mechanisms underlying these effects are not fully elucidated.

The study focuses on four common edible seaweeds in Taiwan, which may limit the generalizability of the findings to other seaweed species or geographical regions. Are there seaweeds produced by commercial growers?  As seaweed composition can vary based on factors such as species, season, and environment, which may influence the bioactivity of seaweed extracts and compounds. If yes, provide details of the company according to the journal format. I was not able to find via google, Everyone Excellent-Algae company.

The study does not provide details on the dose-response relationship of Cl80 or the duration of treatment required for optimal anticancer effects, which are important considerations for translation.

The study does not address potential biological variability in response to Cl80 treatment, which could impact its efficacy.

I would suggest adding a sub section to address the limitations of the study as it is important to note that this study is not conducted in in vivo is crucial for translation.

Author Response

The study on the anticancer properties of seaweed extracts, particularly Cl80, for prostate cancer has several limitations:

The study primarily relies on in vitro cell line models (LNCaP, 22Rv1, PC-3) to assess the anticancer properties of Cl80. While these models provide valuable insights, they may not fully replicate the complexity of human prostate cancer in vivo. It is not clear on what basis authors have selected Cl80 alone for testing.

Response: Thank you for your comment. We have added the reason for testing the anti-PCa cell activity of Cl80 alone in the Results. Additionally, while we acknowledge the limitations of in vitro cell line models, they are commonly used as initial screening tools in cancer research due to their practicality and cost-effectiveness. We agree that further validation in animal models and clinical studies is necessary to corroborate the present in vitro experimental results and determine the potential therapeutic effect of Cl80 on human prostate cancer.

While the study demonstrates the efficacy of Cl80 in inhibiting AR activity, inducing apoptosis, and inhibiting cell proliferation in prostate cancer cell lines, the exact molecular mechanisms underlying these effects are not fully elucidated.

Response: Thanks for your comment. This study started by inhibiting AR activity because AR plays a very important role in the development and treatment of PCa. Therefore, we used bioassay-guided fractionation to obtain Cl80 with AR inhibitory activity. Cl80 significantly inhibits LNCaP cells, which are androgen-dependent PCa cells. It can be seen that Cl80 can affect the growth of PCa by inhibiting AR activity. Interestingly, we did not expect that Cl80 could also inhibit the growth of 22Rv1 and PC-3 cells, which are androgen-independent PCa cell lines. This shows that Cl80 may affect PCa activity through other molecular pathways in addition to the AR pathway. Of course, these hypotheses require further research.

The study focuses on four common edible seaweeds in Taiwan, which may limit the generalizability of the findings to other seaweed species or geographical regions. Are there seaweeds produced by commercial growers?  As seaweed composition can vary based on factors such as species, season, and environment, which may influence the bioactivity of seaweed extracts and compounds. If yes, provide details of the company according to the journal format. I was not able to find via google, Everyone Excellent-Algae company.

Response: Everyone Excellent-Algae company is a traditional small enterprise in Penghu (Pescadores), Taiwan. He does not have a website, his contact number is 886-6-9219757, and his contact address is: No. 578, Liuhe Rd., Xiwei Vil., Magong City, Penghu County 880007, Taiwan (ROC). The company itself is a consolidator and also engages in seaweed farming. The production of M. nitidum and S. cristaefolium is large in the wild and has no economic value in breeding, so the samples used in this study were collected. P. dentata and C. lentillifera have small populations in the wild, so the samples used in this study are cultivated algae.

The study does not provide details on the dose-response relationship of Cl80 or the duration of treatment required for optimal anticancer effects, which are important considerations for translation.

Response: The study does lack specific information on the Cl80 dose-response relationship and the optimal duration of treatment required to achieve optimal anticancer effects. These details, which are critical to the translational aspects of research, will require progressive animal and even human trials to be understood. These discussions will become the next research topics in the future.

The study does not address potential biological variability in response to Cl80 treatment, which could impact its efficacy.

Response: We appreciate your valuable feedback regarding potential biological variability in Cl80 treatment. This is indeed an important aspect to consider in any study involving experimental treatments. Although our study focused on evaluating the effects of Cl80 on prostate cancer cell lines, we acknowledge that biological variability between individual cells and patient samples may impact treatment efficacy. Studies have even pointed out that variability in patient response to even the most effective and targeted treatments is currently a major challenge in drug discovery and patient care, especially in oncology.

I would suggest adding a sub section to address the limitations of the study as it is important to note that this study is not conducted in in vivo is crucial for translation.

Response: We have included a subsection in Discussion to address the limitations of this study. We thank you for bringing this important consideration to our attention and will take it into account in our future research endeavors.

Reviewer 2 Report

Comments and Suggestions for Authors

1. Please write the botanist authority after the scientific name of Alga, for example with Caulerpa lentillifera can be written as Caulerpa lentillifera J.Agardh etc.

2. The introduction shortly mentions previous research on seaweed medicinal properties, but it could benefit from a more detailed review of existing literature related to seaweed's anticancer activity. This will establish a stronger foundation for the current study and highlight its novelty.

3. Please correct the name "Porphyra dentate" to "Porphyra dentata". The computer automatically changed the word "dentata" to "dentate", so please replace all instances of "dentate" with "dentata".

4. You have made a discussion on the C180, it is not clear if C180 is a fraction of a pure compound, if it is a fraction, what is the nature of this fraction? "What are the components of this fraction"?

5. The extraction and fractionation procedure lacks clarity. Can you please provide a more detailed explanation of this process? You mentioned performing elution using ethanol concentrations of 20%, 40%, 60%, 80%, or 100% (v/v). How many fractions were collected from each alga extract, and what was the volume of eluent solvent for each fraction? Additionally, could you specify the method of detection used to collect the fractions (e.g., TLC, detector, spray reagent)

6.  suggest future directions: Including a brief section on future research suggestions. This could involve suggesting potential follow-up studies to further investigate the mechanisms of Cl80 action, its efficacy in animal models, or its potential synergistic effects with existing prostate cancer treatments.

Author Response

1. Please write the botanist authority after the scientific name of Alga, for example with Caulerpa lentillifera can be written as Caulerpa lentillifera J.Agardh etc.

Response: We have completed the requested changes and added botanist authority after the scientific name of the algae.

2. The introduction shortly mentions previous research on seaweed medicinal properties, but it could benefit from a more detailed review of existing literature related to seaweed's anticancer activity. This will establish a stronger foundation for the current study and highlight its novelty.

Response: We have revised the Introduction to provide a more detailed review of the existing literature related to the anticancer activity of seaweeds. We appreciate your suggestions and believe these changes will improve the overall quality and impact of our research.

3. Please correct the name "Porphyra dentate" to "Porphyra dentata". The computer automatically changed the word "dentata" to "dentate", so please replace all instances of "dentate" with "dentata".

Response: Thank you for bringing this to our attention. We have made the necessary corrections, replacing all instances of "Porphyra dentate" with "Porphyra dentata" throughout the document.

4. You have made a discussion on the C180, it is not clear if C180 is a fraction of a pure compound, if it is a fraction, what is the nature of this fraction? "What are the components of this fraction"?

Response: Thank you for your inquiry. We appreciate your interest in our study. To clarify, Cl80 is not a pure compound, but a mixture. We have used Folin & Ciocalteau reagent to analyze the phenolic content of various fractions. The results show that most of the phenolic compounds are distributed between Cl40 and Cl60, among which the phenolic content of Cl80 is only half of that of Cl60. Going forward, we plan to further purify the active components of Cl80 using HPLC, followed by structural analysis using MS and NMR techniques. This will allow us to elucidate the nature of the components in Cl80 and better understand its biological activity.

5. The extraction and fractionation procedure lacks clarity. Can you please provide a more detailed explanation of this process? You mentioned performing elution using ethanol concentrations of 20%, 40%, 60%, 80%, or 100% (v/v). How many fractions were collected from each alga extract, and what was the volume of eluent solvent for each fraction? Additionally, could you specify the method of detection used to collect the fractions (e.g., TLC, detector, spray reagent)

Response: Thank you for your feedback. We have modified the methods section to provide a more detailed procedure. The fractionation procedure we use is simple. Only 5 bed volumes of different concentrations of ethanol were eluted and the 5 bed volumes were collected together. Fractions were concentrated and lyophilized. No detector is used in the fractionation procedure.

6. Suggest future directions: Including a brief section on future research suggestions. This could involve suggesting potential follow-up studies to further investigate the mechanisms of Cl80 action, its efficacy in animal models, or its potential synergistic effects with existing prostate cancer treatments.

Response: Thank you for your suggestion. We include a section on future research directions in the Discussion. We believe that these future research suggestions will help to guide further exploration into the therapeutic potential of Cl80 and contribute to advancing our understanding of its anti-cancer properties.

Reviewer 3 Report

Comments and Suggestions for Authors

Dear authors,

Overall, the manuscript is complex and contain new valuable and interesting information.

Some negative aspects, which I believe need to be corrected in order to publish this manuscript, were also identified.

Abstract

The objectives of the research and background are explained in detail, however when it comes to the results, I believe there should be more information included because the findings are just too general.

Introduction

Some additional data are required.

Line 49 – More specific details from the literature about the overall composition and, in particular, the classes of substances known as bioactive compounds which are responsible for the anti-inflammatory, antibacterial, anticancer, and other activities.

Line 60 – I would prefer the study's objectives to be stated much more explicitly and comprehensively.

Results

The statistical results that were obtained are not shown for Figures 1 and 2. Where do statistical differences show up and what are the p-values?

What exactly do the letters a, b, c, d, and e mean in Figures 3-7?

Figure 5 and Figure 8 - Although images are interesting they also have a visual aid issue because they are far too small. All staining images are sub-standard. It is suggested to enhance those and include high resolution images.

Final note - Some small language and/ or proofing errors should be made.

Author Response

Overall, the manuscript is complex and contain new valuable and interesting information.

Some negative aspects, which I believe need to be corrected in order to publish this manuscript, were also identified.

Abstract

The objectives of the research and background are explained in detail, however when it comes to the results, I believe there should be more information included because the findings are just too general.

Response: We appreciate your positive comments regarding the complexity of the manuscript and the value of the information provided. As suggested, we have modified the results section in the abstract to provide more complete findings.

Introduction

Some additional data are required.

Line 49 – More specific details from the literature about the overall composition and, in particular, the classes of substances known as bioactive compounds which are responsible for the anti-inflammatory, antibacterial, anticancer, and other activities.

Response: Thank you for your suggestion. We revised the manuscript to include more details from the literature regarding the composition of seaweed and its physiological activities, particularly anticancer properties.

Line 60 – I would prefer the study's objectives to be stated much more explicitly and comprehensively.

Response: Thank you for your feedback. We have revised the manuscript to clearly describe the objectives of this study.

Results

The statistical results that were obtained are not shown for Figures 1 and 2. Where do statistical differences show up and what are the p-values?

Response: Thank you for your comment. We have revised the manuscript to include the statistical results for Figures 1 and 2.

What exactly do the letters a, b, c, d, and e mean in Figures 3-7?

Response: In Figure, different letters "a", "b", "c", "d" and "e" represent statistically significant differences (p < 0.05). It will also be noted in the figure legend. Thank you for your feedback.

Figure 5 and Figure 8 - Although images are interesting they also have a visual aid issue because they are far too small. All staining images are sub-standard. It is suggested to enhance those and include high resolution images.

Response: The image in Figure 5 is a representation of a single well (24 well plate). The color of the dye is presented in gray scale, and the higher the color, the greater the number of cells. Because some cells have stagnated growth, they cannot be seen without the microscope. That’s why we dissolved the dye from the well and analyzed it by spectrophotometry. Figure 6 is the field of view of a phase contrast microscope (without staining), the purpose is to observe the closeness of the 500 μm wide gap. Therefore, dotted lines are used as auxiliary marks to allow readers to easily see the cell migration situation. In order to present the degree of migration scientifically, we use ImageJ software to quantify it and create a bar chart.

Final note - Some small language and/ or proofing errors should be made.

Response: Thank you for your feedback. We have carefully reviewed the manuscript and made the necessary language and proofreading edits to ensure clarity and accuracy.

Round 2

Reviewer 2 Report

Comments and Suggestions for Authors

Please state in your manuscript that C180 is a fraction 

Reviewer 3 Report

Comments and Suggestions for Authors

I read the manuscript and noticed that the authors made the requested changes. The quality of the article resubmitted has increased.